# Glycerol Electro-Oxidation in Alkaline Media and Alkaline Direct Glycerol Fuel Cells

**Ermete Antolini** 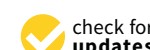

Scuola di Scienza dei Materiali, Via 25 aprile 22, Cogoleto, 16016 Genova, Italy; ermantol@libero.it

**Abstract:** The low price, highly active triol structure, high volumetric energy density, simple storage and environment-friendly properties make glycerol a promising fuel for an alkaline direct alcohol fuel cell (ADAFC). Unlike other ADAFCs, alkaline direct glycerol fuel cells (ADGFCs) can be used either to generate only energy (the common use of fuel cells) or to produce both energy and valuable chemicals. This work presents an overview of catalysts for glycerol oxidation in alkaline media, and their use in ADGFCs. A particular attention was paid to binary and ternary catalysts able both to increase the selectivity to valuable C3 glycerol oxidation products, reducing the C–C bond cleavage, and simultaneously to enhance glycerol conversion.

**Keywords:** glycerol; alkaline direct glycerol fuel cells; electrocatalyst; Pt-, Pd- and Au-based materials

## 1. Introduction

Glycerol (GLY) is a byproduct of biodiesel, and also, in a lower amount, of bioethanol production [1]. Thus, a serious problem in biodiesel production is the consequent production of glycerol as a by-product with a low commercial value. Different methods have been developed to transform glycerol into value-added chemicals [2,3]. Among the products of glycerol oxidation, almost all C3 oxidation compounds are of high commercial value. The oxidation of primary hydroxy groups yields glyceraldehyde (GALD) and glyceric acid (GLA, oxidation of only one primary hydroxy group) and tartronic acid (TA, oxidation of both primary hydroxy groups), which are commercially useful compounds. The oxidation of the secondary hydroxy group yields the fine chemical dihydroxyacetone (DHA), whereas the oxidation of all three hydroxy groups affords the highly functionalized hydroxypyruvic acid (HPA) and mesoxalic acid (MOA) of high commercial value [3,4]. However, these glycerol-derived value-added compounds are mostly produced through either costly and non-environment-friendly oxidation processes, by using strong oxidants, such as chromic acid [5] or slow microbial processes with low product yields [6]. Thus, research efforts have been addressed to glycerol oxidation by "green" and fast heterogeneous catalysis using molecular oxygen [7]. However, the catalytic reactions carried out at 50–60 °C pressurized with oxygen [8,9] can be disadvantageous for the selectivity. Thus, electro-catalytic oxidation of glycerol has been investigated. Koper et al. [10,11] demonstrated that the electrode potential can is a more facile and controlled driven force than oxygen for the catalytic oxidation of glycerol. The use of glycerol byproducts as valuable chemicals, however, is not enough to cover the increased glycerol production, owing to the increased biodiesel fuel production. A way to dispose of some of the large glycerol production is its application as a fuel for an alkaline direct alcohol fuel cell (ADAFC) [12]. The low price, highly active triol structure, high volumetric energy density, simple storage and environment-friendly properties make glycerol a promising fuel for ADAFCs. The average price of crude glycerol is lower than that of the commonly used fuel for ADAFCS, such as methanol (ca. 40% lower) and ethanol (ca. 75% lower.), and also ca. 80% lower that of high purity glycerol [13]. Unlike other fuel cells, alkaline direct glycerol fuel cells

(ADGFCs) can be used either to generate only energy (the common use of fuel cells) or to produce both energy and valuable chemicals. Most parts of works on ADGFCs have been addressed to fully oxidization of glycerol to achieve a high power density, while few investigations have been devoted to a selective (partial) glycerol oxidation to produce valuable chemicals and simultaneously to generate energy. Depending on the type of fuel cell utilization, and, as a consequence, on the degree of glycerol oxidation, different catalysts have to be employed. This work presents an overview of catalysts for glycerol oxidation in alkaline media, and their use in ADGFCs. A particular attention was paid to binary and ternary catalysts able both to increase the selectivity to C3 oxidation products, reducing the C–C bond cleavage and simultaneously to enhance glycerol conversion. The effect of the use of either high purity or crude glycerol on fuel cell performance was highlighted.

## 2. Glycerol Electro-Oxidation in Alkaline Media

While in acidic media the electro-oxidation of glycerol was almost carried out on platinum and platinum-based catalysts, due to the low stability and electrocatalytic activity of other precious metal catalysts, in alkaline media also palladium and gold showed an appreciable activity for the glycerol oxidation reaction (GOR), even if platinum remains the best catalyst for potentials lower than 0.8 V vs. RHE, which are of fuel cell interest. The complete oxidation of glycerol to carbonate in alkaline media produces 14 electrons and is described by the following equation:

$$CH_2OH\text{-}CHOH\text{-}CH_2OH + 20\,OH^- \rightarrow 3\,CO_3^{2-} + 14\,H_2O + 14\,e^-, \tag{1}$$

Equation (1) shows that for the complete oxidation of glycerol 20 moles of $OH^-$ are required. In alkaline media the kinetics of alcohol electro-oxidation is faster than in acidic ones [14], as the base catalyzes the first deprotonation step for the electro-oxidation of alcohols to form alkoxide species, while the second deprotonation step depends on the catalyst [15,16]. Moreover, also non-noble metals, such as nickel-based catalysts, which are not stable in acidic media, presented a not negligible GOR activity in alkaline media.

### 2.1. Precious Metal Catalysts

### 2.1.1. Pure Pt, Pd and Au Catalysts and Binary PtAu and PdAu Catalysts

Generally, on Pt primary OH are easy oxidized, with the formation of GALD, GLA and TA, while the secondary OH are hardly oxidized at low potentials; the products of the oxidation of secondary OH, that is, DHA, HPA and MOA, can be observed only at high potentials [10,17,18]. Glycolic acid (GA), oxalic acid (OA) and formic acid (FA) are also detected indicating the breaking of the C–C bond [10,17,18]. All these acids are present in the base form. The reaction pathway for the glycerol electrooxidation in alkaline medium with C–C bond cleavage and in the absence of the oxidation of the secondary OH is shown in Figure 1 [2,10,19–24]. Various reaction pathways were proposed for GLY oxidation to C2/C1 compounds (see Figure 1): in most of these pathways, GLA is the C3 compound that undergoes the breaking of one of two C–C bonds [10,19–23], followed by TA [2,20,24], MOA [2,19] and also GLY [2], to form C2 and C1 compounds, while C2 compounds, such as GA [20], Glyoxylate (GOA) [21,24] and OA [24], undergo C–C bond cleavage to form FA and carbonate. Sandrini et al. [25] reported that the GOR mechanism on Pt begins with glycerol dehydrogenation to alkoxides followed by the formation of aldehyde intermediates adsorbed as $\eta^1$(O)-aldehyde and $\eta^1$(C)-acyl geometry, respectively. For potential higher than 0.9 V vs. RHE Pt suffers strong deactivation, due to the formation of an inhibiting surface oxide [10,17,18]. Glycerol oxidation on Pd lead to a selectivity towards GALD and GLA at low potentials, and then towards HPA, dicarboxylates and products derived by C–C bond cleavage (C2 and C1 compounds) at higher potentials [18,26]. It has to be denoted that the structure and morphology of the catalysts can play a role for the selectivity: GLA and TA were obtained on Pd/C, and HPA on Pd nanofoam [2].

**Figure 1.** Representation of the possible pathways of GLY electrooxidation on precious metal and precious metal-based catalysts in an alkaline medium with C–C bond cleavage and in the absence of the oxidation of the secondary OH, based to the pathways reported in various schemes [2,10,19–24].

Generally, while platinum and palladium seem to favor the activation and oxidation of primary alcohol functions toward GALD and further toward carboxylates (GLA and TA), on Au surfaces the secondary OH can be more easily activated, leading to DHA formation. Then, the oxidation of primary OHs in DHA lead to the formation of HPA and MOA, and, through a different pathway, lactate (LA) [18–20,27,28]. DHA was also detected on Pt/C and Pd/C by Simoes et al. [18], but no HPA: they proposed that the isomerization equilibrium between DHA and GALD could be displaced toward the formation of the aldehyde on the Pt/C and Pd/C catalysts comparatively to the Au/C catalyst, hindering the formation of HPA. C–C bond cleavage also occurs on Au. The reaction pathway for the glycerol electrooxidation in alkaline medium with the oxidation of primary and secondary OHs and in the absence of C–C bond cleavage is shown in Figure 2 [18–21,28,29]. Partial oxidation of GLY to MOA, the more deeply oxidized C3 compound, without C–C bond breaking, leads to 10 electrons generation against 14 for the complete GLY oxidation to $CO_3^{2-}$. The electrooxidation of glycerol on Au in alkaline media is represented by a complex mechanism with the formation of GALD, GLA, TA, DHA, HPA, LA, MOA, GA, FA and carbonate [19,20,27,28,30]. Moreover, on Au the formation of carbonate seems to be caused by the entrance of oxygen in the intact molecule or by the previous formation of formate radical, without CO formation [30]. For this reason, the stability (poisoning tolerance) of glycerol oxidation on the Au electrode was higher than that of glycerol oxidation on Pt and Pd electrodes [31].

On gold, compared to the oxidation of other low molecular weight alcohols, such as methanol, ethanol, n-propanol, isopropanol and ethylene glycol, glycerol showed the highest activity in alkaline medium [31]. To better understand the electro-oxidation of glycerol on gold, it was compared with that of similar C3 alcohols, that is, 1-propanol, 2-propanol, propane-1,2-diol and propane-1,3-diol [32]. The reactivity of these alcohols on Au in alkaline medium decreased in the following order: glycerol > propane-1,2-diol ≫ propane-1,3-diol > 2-propanol ≈ 1-propanol. For glycerol and propane-1,2-diol, the presence of C2 products indicated a C–C bond cleavage. On the other hand, the electrochemical oxidation of propane-1,3-diol, 2-propanol and 1-propanol led only to 3C products. Based on these results, it was inferred that the presence of vicinal OH groups in the alcohol structure should be a key factor for the C–C bond cleavage in alkaline medium.

**Figure 2.** Representation of the possible pathways of glycerol (GLY) electro-oxidation on precious metal and precious metal-based catalysts in an alkaline medium with the oxidation of primary and secondary OH and in the absence of C–C bond cleavage, based on the pathways reported in various schemes [18–21,28,29].

A comparison of the catalytic activity toward glycerol electro-oxidation in alkaline media of Pt, Pd and Au was carried out in some papers [18,33–35]. In all cases the onset potential for Gly oxidation was in the order Pt > Pd > Au. In terms of the maximum current density, instead, the most active metal was Au. However, on gold high overpotentials are required to obtain an appreciable activity for glycerol electro-oxidation. Apart from the poor GOR activity at low potentials, gold presents some advantages over palladium or platinum, that is, it is more electrochemically stable in alkaline media, is more abundant and it has a lower price than the other precious metals [36]. Moreover, among Pt, Pd and Au electrodes, in alkaline media Au has the lowest activation energy for the electro-oxidation of glycerol [37]. Finally, Au has a very high degree of resistance to the formation of poisoning oxides such as CO [38]. For these reasons and to enhance the GOR activity and short term electrochemical stability, the effect of Au addition to Pt [20,29,39–42] and Pd [18,40,43–50] catalysts on their electrochemical properties in alkaline media was investigated. The GOR activity, in terms of both onset potential and peak current density, and the poisoning tolerance of all PtAu and PdAu catalysts was higher than that of bare Pt and Pd, as can be seen in Table 1. The optimum Au content was in a wide range from ca. 25 Au% [40] to 70%–85 Au% [29], this difference depending on various factors, internal, such as the alloying degree and surface composition, and external, such as temperature and OH$^-$ concentration.

For example, as can be seen in Figure 3, where the PtAu current density to the Pt current density ratio is plotted against Au content for datasets from Li et al. [40] and Zhou et al. [20], in both cases the current density of PtAu at the maximum was ca. 4.5 times that that of pure Pt, but for the data from Li et al. [40] the maximum was at ca. 25 Au%, while for the data from Zhou et al. [20] the maximum was at 65 Au%. The higher GOR activity of Au-containing catalysts than pure metals was ascribed mainly to the bifunctional effect [20,40,41,43–45,48,50] but also to electronic effects [29,39,41,47–50] and the higher electrochemically active surface area (ECSA) [39,40]. According to Mougenot et al. [43], the PdAu surface alloy composition has a negligible effect on the catalytic activity, but the improvement of the GOR activity is due to the presence of non-alloyed gold sites on the catalyst surface, where GLY is adsorbed on Pd atoms and OH- species are adsorbed on Au atoms, allowing the oxidation of GLY oxidation intermediate species at low potentials by the bifunctional mechanism. The enhancement of the GOR activity by Au presence was also ascribed to electronic effects, related to the charge transfer from Pt and Pd to Au, due to the higher electronegativity of Au (2.54) than Pd (2.20) and Pt (2.28). The introduction of Au atoms in Pt lattice could modulate the binding strength of adsorbates on Pt surface: the downshift of Pt d-band center relative to the Fermi level should weak the Pt-GLY oxidation

intermediate species bonding, making easier GLY oxidation intermediate species desorption from Pt surface and making Pt sites available to the adsorption of GLY and $OH^-$ on Pt sites, thus improving glycerol oxidation.

**Table 1.** Glycerol oxidation reaction (GOR) activity and selectivity in alkaline media of binary PtAu and PdAu catalysts.

| Catalyst | GOR Activity | Selectivity (E,V) | Reference |
|---|---|---|---|
| $Pt_{1-x}Au_x$@Ag, x: 0.3–0.8 | $Pt_{1-x}Au_x$ > Pt/C, max: x = 0.60 | C3 (0.5, 0.7, 0.9 and 1.3 V vs. RHE) | [20] |
| $Pt_{1-x}Au_x$, x: 0.10–0.75 | $Pt_{1-x}Au_x$ > Pt, max: x = 0.35 | - | [39] |
| $Pt_{1-x}Au_x$, x: 0.10–0.75 | $Pt_{1-x}Au_x$ > Pt, max: x = 0.85 | LA (0.45–0.6 V vs. RHE) | [29] |
| $Pt_{1-x}Au_x$/C, x: 0.15–0.33 | $Pt_{1-x}Au_x$/C > Pt/C, max: x = 0.20 | - | [40] |
| $Pt_{1-x}Au_x$ /C, x: 0.10–0.50 | $Pt_{1-x}Au_x$/C > Pt/C, max: x = 0.50 | - | [41] |
| PtAu/C | PtAu/C Pt/C | - | [42] |
| $Pd_{1-x}Au_x$/C, x: 0.15–0.33 | $Pd_{1-x}Au_x$/C > Pd/C, max: x = 0.20 | - | [40] |
| $Pd_{1-x}Au_x$, x: 0.5, 0.7 | $Pd_{1-x}Au_x$ > Pd, max: x = 0.50 | - | [18] |
| $Pd_{0.7}Au_{0.3}$ | $Pd_{0.7}Au_{0.3}$ > Pd | - | [43] |
| $Pd_{0.5}Au_{0.5}$/C | $Pd_{0.5}Au_{0.5}$/C > Pd/C | - | [44] |
| $Pd_{1-x}Au_x$/C, x: 0.25, 0.5, 0.75 | $Pd_{1-x}Au_x$/C > Pd/C, max: x = 0.50 | C2,C1 (−0.4–0.05 V vs. Ag/AgCl) | [45] |
| $Pd_{1-x}Au_x$/VGNCF, x: 0.25, 0.33, 0.50 | $Pd_{1-x}Au_x$/VGNCF > Pd/VGCNF, max = 0.50 | - | [47] |
| PdAu/NPSS | PdAu/NPSS > Pd/NPSS | - | [48] |
| $Pd_{1-x}Au_x$/P-Se-C, x: 0.25, 0.33, 0.50 | $Pd_{1-x}Au_x$/P-Se-C > Pd/P-Se-C, max = 0.50 | - | [49] |
| $Pd_{1-x}Au_x$/C, x: 0.25, 0.5 | $Pd_{1-x}Au_x$/C > Pd/C, max: x = 0.50 | - | [50] |

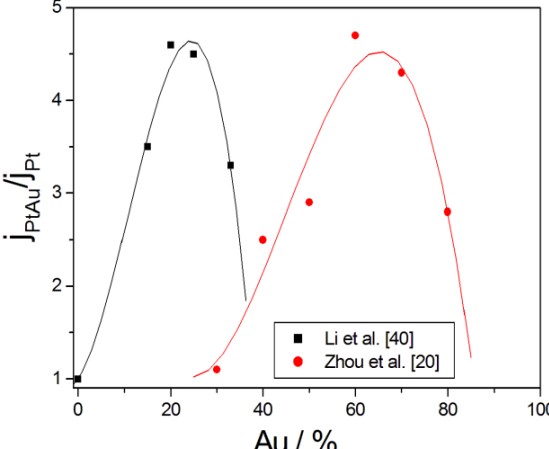

**Figure 3.** Dependence of the PtAu current density to Pt current density ratio on Au content in the catalyst for datasets from Li et al. [40] and Zhou et al. [20].

Regarding the products selectivity, the C3/C2 ratio of GLY oxidation products is of great interest, being indicative of the selectivity of the material. The higher the C3/C2 ratio, the higher the selectivity to the high valuable C3 products. Zhou et al. [20] evaluated the products of GLY oxidation on $Pt_xAu_y$ catalysts prepared by using Ag nanoparticles as sacrificial seeds. As can be seen in Figure 4, the addition of Au remarkably increased the C3 selectivity of Pt at 0.5 V vs. RHE: the highest C3/C2 value was shown by the $Pt_5Au_5$@Ag catalyst. Among C3 products, the highest DHA selectivity (77.1%) was shown by $Pt_4Au_6$@Ag at 1.1 V vs. RHE, the highest GALD selectivity (35.0%) by $Pt_6Au_4$@Ag at 1.1 V vs. RHE, and the highest GLA selectivity (34.5%) by $Pt_3Au_7$@Ag at 0.7 V vs. RHE. Generally, the selectivity of C3 products decreases for high Au contents (>60 Au%), particularly for potentials >0.7 V vs. RHE. An increase of C3 selectivity with increasing Au content in PtAu catalysts was observed also for GLY

oxidation in a base free aqueous solution [51]. Dai et al. [29] reported a higher selectivity for lactic acid on PtAu catalysts than on bare Pt. The highest lactic acid selectivity and glycerol conversion were observed on Pt-enriched PtAu catalysts.

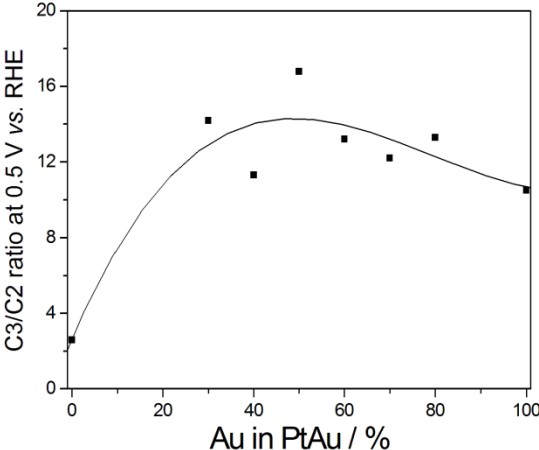

**Figure 4.** Dependence of the C3/C2 ratio of GLY oxidation products at 0.5 V vs. RHE on Au content in PtAu catalysts from data in ref. [20].

### 2.1.2. Pt- and Pd- and Au- Based Binary/Ternary Catalysts

The effect of the addition of d-group (transition metals) and p-group (post transition metals) elements, in an alloyed or non-alloyed oxide form, to Pt, Pd and Au catalysts was investigated to aim to improve their electrochemical performance and/or selectivity. Generally, the addition of a second/third metal increases both the GOR activity, in terms of both lower onset potential and higher oxidation peak current, and the stability of Pt, Pd and Au catalysts. The increase of the mass activity is due to the enhancement of the specific activity and/or the electrochemically active surface area. The comparison of the GOR activity of binary/ternary catalysts with that of the respective pure metal catalysts and their selectivity are reported in Table 2 [13,18,24,26,28,33,44,50,52–88]. Generally, the addition of a second/third metal improves the GOR activity, and an optimum catalyst composition is observed. As for Au-containing binary catalysts previously reported, the higher GOR activity of binary/ternary catalysts than pure metals was ascribed to the bifunctional effect and/or electronic effects. According to the bifunctional mechanism, GLY is adsorbed on Pt, Pd or Au atoms and $OH^-$ species are adsorbed on second metal atoms, allowing the oxidation of adsorbed intermediate species at low potentials. On the other hand, electronic effects by alloying a second metal to Pt can result in a downshift in the d-band center of Pt, reducing the adsorption energy and binding strength of GLY oxidation intermediates on the Pt surface, facilitating their desorption from Pt surface and making Pt sites available to the adsorption of GLY and $OH^-$ on Pt sites, thus improving glycerol oxidation. For d-group elements the increase of the GOR activity can be due to an increase of GLY conversion, without a change of the product distribution, and/or an increase of the GLY oxidation degree, either by increasing the amount of more deeply oxidized C3 compounds (GLA, TA and MOA), as in the case of PtNi [21,52], without a change of the C3/C2 ratio, or by C–C bond cleavage, forming C2 and C1 compounds, as for PtRh [52], PtAg [56] and PdAg [13], decreasing the C3/C2 ratio. Both an increase of more deeply oxidized C3 compounds and C–C bond breaking was observed for GLY oxidation on $PtNiOH_2/C$ by Velazquez-Hernandez [21], whereas an increase of C3 products and a decrease of C2 compounds occurred by GLY oxidation on PtRu [52].

**Table 2.** GOR activity and selectivity in alkaline media of binary and ternary Pt-, Pd- and Au-based catalysts.

| Catalyst | GOR Activity | Selectivity (E,V) | Reference |
|---|---|---|---|
| PtRh/GNS, PtRhNi/GNS | PtRh/GNS, PtRhNi/GNS > Pt/GNS | C2 (−0.4 V vs. SCE) | [52] |
| PtNi/GNS | PtNi/GNS > Pt/GNS | TA (−0.4 V vs. SCE) | [52] |
| PtNi(OH)$_2$/C | PtNi(OH)$_2$/C > Pt/C | MOA,GA,FA (−011 V to 0.43 V vs. NHE) | [21] |
| PtNiO/Ti | Pt$_{0.8}$Ni$_{0.2}$O/Ti > Pt/Ti, Pt$_{0.5}$Ni$_{0.5}$O/Ti, Pt$_{0.2}$Ni$_{0.8}$O/Ti < Pt/Ti, | - | [53] |
| PtCo/CNT | PtCo/CNT > Pt/CNT | - | [54] |
| PtRu/GNS | PtRu/GNS > Pt/GNS | C3 (−0.4 V vs. SCE) | [52] |
| PtCu | Pt$_1$Cu$_1$, Pt$_7$Cu$_3$ > Pt, Pt$_9$Cu$_1$ < Pt | - | [55] |
| PtAg/C, PtAg/MnO$_x$/C | PtAg/MnO$_x$/C > PtAg/C > Pt/C | C1 (0.5–0.6 V vs. RHE) | [56] |
| Pt$_9$Bi$_1$/C | Pt$_9$Bi$_1$/C > Pt/C | - | [57] |
| PtBi | PtBi > Pt | C3 (0.25–0.60 V vs. RHE) | [58] |
| Pt$_9$Bi$_1$/C | Pt$_9$Bi$_1$/C > Pt/C | C3 (0.3–0.7 V vs. RHE) | [59] |
| PtBi | PtBi > Pt | GLA (0.7–0.85 V vs. RHE) | [60] |
| PtCeO$_2$/C | PtCeO$_2$/C > Pt/C | - | [61] |
| PtCeO$_2$/C | PtCeO$_2$/C > Pt/C | C3 (−0.4 V vs. SCE) | [22] |
| PtCuCo | PtCuCo > PtCu > PtCo > Pt/C | - | [62] |
| PtCoNi/C | PtCoNi/C > PtCo/C, PtNi/C > Pt/C | - | [63] |
| PdRh | PdRh > Pd | CO$_3$$^{2-}$ (0.3–0.9 V vs. RHE) | [64] |
| Pd$_3$Ru/NC | Pd$_3$Ru/NC > Pd/NC | - | [65] |
| Pd$_3$Ag$_1$/C, Pd$_1$Ag$_1$/C | Pd$_3$Ag$_1$/C, Pd$_1$Ag$_1$/C >Pd/C | - | [50] |
| Pd$_9$Cu$_1$/C | Pd$_9$Cu$_1$/C >Pd/C | - | [66] |
| Pd$_3$Cu/NMC | Pd$_3$Cu/NMC/Pd/C | - | [67] |
| PdCu/C | PdCu/C > Pd/C | - | [68] |
| PdAg, PdAg$_3$ | PdAg, PdAg$_3$ > Pd/C | C2 (n.d.) | [13] |
| PdCo/Au | PdCo/Au > Pd/Au | - | [69] |
| PdMo/C | PdMo/C > Pd/C | - | [70] |
| PdNi/C | PdNi/C > Pd/C | - | [18] |
| PdNi/C | PdNi/C > Pd/C | - | [71] |
| Pd$_9$Bi$_1$/C | Pt/C = Pd$_9$Bi$_1$/C > Pd/C | - | [57] |
| PdBi | PdBi > Pd | - | [58] |
| PdBi | PdBi > Pd | DHA (0.35–0.7 V vs. RHE) CO$_3$$^{2-}$, HPA (0.7–1.0 V vs. RHE) | [26] |
| Pd$_1$Sn$_1$, Pd$_1$Sn$_2$, Pd$_1$Sn$_3$ | Pd$_1$Sn$_1$, Pd$_1$Sn$_2$ > Pd, Pd$_1$Sn$_3$ < Pd | C3 (0.5–0.9 V vs. RHE) | [72] |
| PdMO$_x$/C, (MO$_x$ = CeO$_2$, NiO, Co$_3$O$_4$, Mn$_3$O$_4$) | PdMxOy > Pd/C | - | [73] |
| PdRuCo | PdRuCo > Pd/C | - | [74] |
| PdAu/C, PdSn/C, PdAuSn/C | PdAu/C = Pd$_5$Au$_4$Sn$_1$/C > Pd$_5$Au$_1$Sn$_4$/C > Pd PdSn/C < Pd/C | - | [44] |
| PdNiOP/C | PdNiOP/C > Pd/C | - | [75] |
| FeCo@Fe@Pd/C | FeCo@Fe@Pd/C > Pd/C | - | [76] |
| PdM (M = Sb, Sn, or Pb) | PdM > Pd | - | [77] |
| PdIr | PdIr lower onset potential and j than Pd | CO$_3$$^{2-}$ (0.5–1.0 V vs. RHE) | [78] |
| AuAg/C | AuAg/C > Au/C | - | [79] |
| AuAg/C | AuAg/C < Au/C | C1 (0.5–1.6 V vs. RHE) | [28] |
| AuAg | AuAg/C > Au/C | C1,C2 (n.d.) | [80] |
| AuAg/C (5–30 wt% Au) | 5, 10 wt% AuAg > Au, 30 wt% AuAg < Au | - | [81] |
| Au/Ni | Au/Ni > Au | - | [82] |
| Au$_2$Ni/C | Au$_2$Ni/C > Au/C | - | [83] |
| Au/Cu/C | n.d. | C3 (GLA,TA; 0.0–0.1 V vs. Ag/AgCl) | [84] |
| AuCu | AuCu > Au | - | [85] |
| AuMnO$_2$/C | AuMnO$_2$(5,9%)/C > Au, AuMnO$_2$(16,23%)/C < Au | - | [86] |
| AuCeO$_2$/C | AuCeO$_2$C > Au/C | - | [87] |
| Au-Co$_3$O$_4$, -NiO, -Mn$_3$O$_4$ and -MgO/C | Au-Co$_3$O$_4$, -NiO, -Mn$_3$O$_4$ and -MgO/C >Au/C | - | [88] |

As evinced by these results, there is no correlation between the GOR activity and the C3/C2 ratio, also if, at a fixed GLY conversion, the current density should increase with decreasing the C3/C2 ratio, that is, by oxidation of C3 to C2 compounds, increasing the number of generated electrons per oxidized GLY molecule. An increase of the current density with decreasing C3/C2 ratio, mostly due to the increase of C2 compound amount by Rh-promoted C–C bond breaking, is visible in Figure 5,

showing the histogram of the current density at −0.4 V vs. SCE for Pt/CNS, binary PtNi/GNS and PtRh/GNS, and ternary PtRhNi/GNS catalysts with different selectivity, from the data of Zhou et al. [52]. Among the different metals, Ru [24,52] and Cu [84] increase the selectivity to GLY and TA, and Ni to TA [52] and MOA [21]. Different effects on C–C bond cleavage of d-group elements have been reported: Rh [13,52], Ag [13,28,80] and Ni [21] facilitate C–C bond breaking, but Ru [52] and Cu [83] reduce and suppress, respectively, the C–C bond cleavage. By density functional theory (DFT) modeling of adsorption of CH, CO and CHCO, the reactivity of Pt and $Pt_3M$ surfaces (M = Rh, Re, Ru, Ni and Sn) for C–C bond cleavage in the simple CHCO moiety present in GLY oxidation intermediate species was studied [89]. Indeed, a relation between the adsorption energies of these molecules and activation energy for C–C bond cleavage was established, that is, the surfaces with higher adsorption energies of CH and CO have lower activation energy for C–C bond breaking. Among Pt and $Pt_3M$ surfaces, $Pt_3Rh$ alloy presented the highest CH and CO adsorption energies, and hence the highest reactivity for C–C bond cleavage. On the other hand, as previously reported, electronic effects by alloying a second metal to Pt can result in a downshift in the d-band center of Pt, reducing the adsorption energy and binding strength of C3 GLY oxidation intermediate species on the Pt surface, making easier their desorption from Pt surface, thus hindering their further decomposition into C2/C1 compounds [22]. The presence of $Au^+$, generated by an interaction between Au and electro-deposited $Cu_2O$, increases the selectivity to GLY and TA and suppressed C–C bond breaking [84]. The addition of p group elements, such as bismuth and Sn, to Pt, Pd and Au catalysts results in a lower glycerol oxidation onset potential and a higher GOR activity, and gives rise to a change in selectivity. The addition of bismuth and tin to platinum leads to reduce/suppress the dissociative adsorption of glycerol (C–C bond breaking) and the formation of adsorbed CO species, enhancing the selectivity toward C3 products [59,60,72]. Bismuth hinders C–C bond cleavage both in acid and alkaline media, but while in acid medium Bi presence results in the formation of DHA with 100% selectivity [90], in alkaline media GLA is the main product, and DHA is either produced in extremely low quantities or not produced [59,60]. Indeed, in acid medium Bi blocks the pathway for primary oxidation and also provides a specific Pt–Bi surface site for secondary alcohol oxidation, but in alkaline medium the $OH^-$ presence suppress the blocking of the pathway for primary oxidation, giving rise to the formation of GALD and GLA. Conversely, on PdBi a selectivity for $CO_2$ and carbonate production, and formation of HPA was observed in alkaline media [26]. Couteanceau et al. [58] compared GLY oxidation on PdBi and PtBi in alkaline media. They suggested that Pd nanoparticles were decorated by $Bi_2O_3$, or more likely $Bi(OH)_3$ clusters. The modification of Pd by Bi does not affect the selectivity of glycerol oxidation, whereas in the case of PtBi a change in selectivity takes place. Electronic effects and/or the ensemble (third body) effect of alloyed Bi in PtBi should give rise to the suppression of C–C bond cleavage, whereas $Bi_2O_3$ and $Bi(OH)_3$ in PdBi should enhance the complete oxidation of GLY to $CO_2$ by the bifunctional effect.

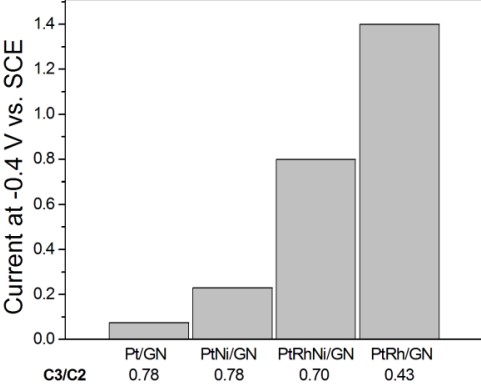

**Figure 5.** Histogram of the current density at −0.4 V vs. SCE for Pt/CNS, binary PtNi/GNS and PtRh/GNS, and ternary PtRhNi/GNS catalysts with different selectivity from the data of Zhou et al. [52].

## 2.2. Non-Precious Metal Catalysts

Notwithstanding precious metal (Pt, Pd and Au) based electrocatalysts are considered the best electrocatalysts for glycerol oxidation in alkaline media, their high prices and limited sources of precious metals, as well as the not too high electrochemical stability and the poor resistance of Pt to the formation of poisoning species are a drawback for large-scale commercial applications. Thus research efforts were addressed to non-precious electrocatalysts. Among non-precious metals, nickel-based electrocatalysts, in addition to low price and large sources, have an anti-poison ability and long-term stability in alkaline solutions, making them attractive candidates for GLY oxidation in alkaline media. It is known that Ni possesses an appreciable activity for alcohol oxidation in alkaline media [91]. Since nickel placed in contact with an alkaline solution becomes covered with a layer of nickel hydroxide, the surface change can be written as:

$$Ni + 2OH^- \rightarrow Ni(OH)_2 + 2e^-. \tag{2}$$

$$Ni(OH)_2 + OH^- \rightarrow NiOOH + H_2O + e^-. \tag{3}$$

Although the oxidation state of the nickel in the oxide layer likely changes continuously between two and three over a range of potentials [92]. NiOOH can exist in two crystal structures, namely β- and γ-forms. In some conditions, β-NiOOH can be converted to γ-NiOOH [93]. The transformation of β-NiOOH to γ-NiOOH is not desirable, as it is accompanied with large volumetric change leading to poor electric contact between the current collector and the catalysts. So, the suitable surface species for glycerol electrooxidation is β-NiOOH. The "indirect" electron transfer mechanism, proposed by Fleischmann et al. [92] can be represented as:

$$\beta\text{-NiOOH} + C_3H_8O_3 \rightarrow \beta\text{-Ni(OH)}_2 + \text{products}. \tag{4}$$

During this reaction the glycerol is oxidized to intermediate compounds and NiOOH is reduced to Ni(OH)$_2$, depleting of NiOOH the electrode surface.

In addition to the indirect electron transfer, a direct mode of electron transfer in alcohol oxidation on Ni in alkaline medium was reported [94]. In this case, glycerol molecules incorporate into the Ni hydroxide surface and are oxidized by surface OH$^-$ ions, without NiOOH consumption, according to the reaction:

$$Ni(OH)_2 + C_3H_8O_3 \rightarrow Ni(OH)_2(C_3H_8O_3)ads \rightarrow NiOOH + \text{product} + H^+ + e^- \tag{5}$$

The mechanism of glycerol oxidation changes from direct at low GLY concentration to indirect at high GLY concentration. The direct mechanism involves a Langmuir-type adsorption process on Ni(OH)$_2$ sites for GLY, followed by hydrogen abstraction from the α carbon to the OH group trough a radical pathway in the rate-determining step, according to the following scheme [95]:

$$RCH_2OH + Ni(OH)_2 \rightarrow Ni(OH)_2(RCH_2OH)_{ads}. \tag{6}$$

$$Ni(OH)_2(RCH_2OH)_{ads} + NiOOH \rightarrow Ni(OH)_2 + Ni(OH)_2(R\bullet CHOH)_{ads} + H_2O. \tag{7}$$

$$Ni(OH)_2(R\bullet CHOH)_{ads} + NiOOH \rightarrow Ni(OH)_2 + Ni(OH)_2(RCOH)_{ads} + H_2O. \tag{8}$$

The glycerol oxidation on Ni in alkaline media was reported in some works [96–101], starting from different Ni structures, such as electrosynthesized nanocrystalline hexagonal close-packed (hcp) nickel [96], sinusoidal-wave electrochemically treated Ni surface [97], a Ni nanoparticle modified boron doped diamond (BDD) electrode [98], a nickel ion implanted-modified indium tin oxide electrode [99], pulse electrodeposited Ni with two structure directing agents, that is, citric acid (CA) and tetrabutylammonium bromide (TBr) [100] and Ni(OH)$_2$ nanoparticles encapsulated with poly[Ni(*salen*) [101]. All these Ni catalysts presented a GOR activity higher than the conventional Ni

electrode, however, the performance of glycerol oxidation was too poor to be used for commercial applications. So, to enhance the GOR activity, the effect of Co, Fe, Cu, Pd and TiO2 addition on the GOR activity of Ni was investigated [102–107]. The starting materials were both NiM (M = Co, Fe, Cu and Pd) alloys [102–104,106] and mixed Ni and Co oxides [105]. The addition of Co, Cu and Pd to Ni improves the catalytic activity of the Ni-based electrocatalysts for glycerol oxidation [102–106]. In the case of NiCo catalysts, independently of the starting material, the reactive redox couples were $Ni(OH)_2/NiOOH$ and $Co(OH)_2/CoOOH$. The mechanism of the glycerol oxidation at the $Co(OH)2/CoOOH$ redox couple is very similar to that at the $Ni(OH)2/NiOOH$ redox couple, but the $Co(OH)2/CoOOH$ redox couple produces a considerably lower current density than that by the $Ni(OH)_2/NiOOH$ redox couple [108]. The positive effect of Co on the GOR activity of Ni was explained in various ways. It was reported that, in the presence of Co, NiOOH is formed at a lower potential [109]. The addition of cobalt to nickel shifts the onset potential and the peak of reaction (3) to lower potential values by the formation of a $Co_xNi_{1-x}(OH)_2/Co_xNi_{1-x}OOH$ redox couple [102–110]. The replacement of a small amount of nickel (0.9 at%) by cobalt decreased the amount of Ni alloyed from 26 to 15 at%, promoting nickel change from metal form into nickel hydroxide form [63]. Finally, DFT calculations showed that the addition of Co to Ni could remarkably enhance the surface coverage of the redox species and weaken the CO adsorption [111]. The presence of $TiO_2$ also modifies chemical state of nickel: the Ni surface phase of $TiO_2$-Ni/C mainly consisted of $Ni(OH)_2$ while that of uncoated Ni/C was a mixed phase of NiO and $Ni(OH)_2$ [106] The $TiO_2$-Ni/C showed a GOR activity 2.4 times higher than that of Ni/C. Regarding the selectivity of Ni-based catalysts, generally, $Ni(OH)_2/NiOOH$ and the other Co and Fe oxides/hydroxides lead to a C–C bond cleavage resulting in the formation of C2 and C1 compounds as the main reaction products. In particular, Co presence in the nickel based catalysts enhances the C–C bond cleavage of GLY.

In addition to nickel, another non-precious catalyst alternative to noble metals for the selective oxidation of alcohols is copper [112]. CuCo [113], Cu/CoO [114,115] and $Cu/Al_2O_3$ [116] couples were investigated as nonprecious metal for GOR oxidation. For CuCo-based materials prepared by co-precipitation, CoO(OH) in contact with CuO was identified as the potentially active phase. Rather than the GOR activity, the selectivity of these catalysts was evaluated: Generally, as in the case of Ni-based catalysts, also for Cu-based catalysts a selectivity to C2 and C1 products was observed.

Glycerol oxidation in alkaline media on cobalt-based catalysts was also investigated [117]. Cobalt catalysts supported on $Mg_3Al(OH)_y(CO_3)_z$ prepared by a two-step modified sol–gel method showed 100% glycerol conversion with 64% and 24% selectivity to TA and OXA, respectively. Finally, based on the Pt-like electrochemical performance of tantalum carbide (TaC), fluorine doped $TaC_xF_yO_z/C$ was investigated as non-precious metal electrocatalysts for glycerol oxidation in alkaline medium [118]. $TaC_xF_yO_z/C$ showed a lower GOR onset potential and a higher exchanged current density, CO tolerance and stability than Pt.

The comparison of the GOR activity with that of the respective standard pure metal catalyst and the selectivity of non-precious metal catalysts are reported in Table 3. With only one exception, no comparison of the GOR activity of non-precious metal catalysts with that of precious metals was made. This may suggest that the GOR activity of non-precious metal catalysts is not comparable, that is, considerably lower, than that of precious metal catalysts.

Summarizing, the research carried out so far was mainly aimed to the improvement of the catalytic activity and selectivity of noble metals by the addition of a second/third metal. Few works, instead, was addressed to the improvement of the activity for glycerol oxidation of less expensive non-precious metals, whose selectivity for C2 and C1 products is high. Therefore, much space remains for research aimed at increasing the catalytic activity of non-precious catalysts, maintaining their high selectivity.

**Table 3.** GOR activity and selectivity in alkaline media of non-precious catalysts.

| Catalyst | GOR Activity | Selectivity (E,V) | Reference |
|---|---|---|---|
| electrochemically treated Ni | Treated Ni > untreated Ni | GALD (0.34–0.54 V vs. Hg/HgO) | [97] |
| Ni-boron doped diamond | Ni-BDD > bulk Ni macro electrode | - | [98] |
| pulse electrodeposited Ni-Ca, Ni-TBr | pulse electrodeposited Ni-Ca, Ni-TBr > bare Ni | - | [100] |
| poly[Ni(*salen*)] encapsulated Ni(OH)$_2$ nanoparticles | - | FA (n.d.) | [101] |
| NiCo/C | NiCo/C > Ni/C | C2,C1 (1.3–1.9 V vs. RHE) | [102] |
| NiCo/C, NiFe/C | NiCo > Ni/C, NiFe/C < Ni/C | C2,C1(1.2–1.9 V vs. RHE) | [103] |
| NiCo/CCE, NiCu/CCE | NiCo/CCE, NiFe/CCE > Ni/CCE | - | [104] |
| NiCoO$_2$ | NiCoO$_2$ > NiO, Co$_3$O$_4$ | - | [105] |
| NiPd | NiPd > Ni | GALD (0.34 V vs. Hg/HgO) | [106] |
| Ni-TiO$_2$/C | Ni-TiO$_2$/C > Ni/C | - | [107] |
| CuCo | - | C2,C1 (n.d.) | [113] |
| Cu/CoO | - | C2,C1 (n.d.) | [114,115] |
| Ordered mesoporous Cu-Al$_2$O$_3$ | Cu-Al$_2$O$_3$ > Al$_2$O$_3$, Ordered Cu-Al$_2$O$_3$ > non-ordered Cu-Al$_2$O$_3$ | C2,C1 (n.d.) | [116] |
| Co/Mg$_3$Al(OH)$_{y}$(CO$_3$)$_z$ | - | TA (64%), OXA (24%) (n.d.) | [117] |
| TaC$_x$F$_y$O$_z$/C | TaC$_x$F$_y$O$_z$/C > Pt | - | [118] |

## 3. Alkaline Direct Glycerol Fuel Cells (ADGFCs)

Alkaline direct alcohol fuel cells (ADAFCs) have attracted special attention, due to the fast electrochemical kinetics, allowing to use non-precious metal catalysts and low catalysts loadings, and the reduced fuel crossover, allowing to use thin membranes, enhancing fuel cell performances [91]. Basically, an ADAFC consists of an anode and a cathode, separated by an alkaline anion-exchange membrane (AAEM). As previously reported, the complete oxidation of glycerol to carbonate in alkaline media can occur at the anode and is described by the following equation:

$$CH_2OH\text{-}CHOH\text{-}CH_2OH + 20\,OH^- \rightarrow 3\,CO_3^{2-} + 14\,H_2O + 14\,e^-,$$

At the cathode, oxygen is reduced with formation of OH$^-$, according to the Equation (9):

$$7/2\,O_2 + 14\,e^- + 7\,H_2O \rightarrow 14\,OH^-, \tag{9}$$

and the overall equation is described by the Equation (10):

$$CH_2OH\text{-}CHOH\text{-}CH_2OH + 7/2\,O_2 + 6\,OH^- \rightarrow 3\,CO_3^{2-} + 7\,H_2O. \tag{10}$$

The presence of OH$^-$ ions in the fuel stream of ADAFCs is mandatory to obtain high performances. When OH$^-$ ions are present in the fuel stream, alkaline fuel cells showed a better performance than the corresponding acidic fuel cells, independently of the type of fuel [119,120]. In the absence of OH$^-$ ions in the fuel stream, the cell performance is very poor. The presence of OH$^-$ ions in the fuel, however, can lead to some problems, such as corrosion, electrode weeping, mechanic electrode destruction by carbonate crystals and catalyst decomposition [121]. So, it is necessary to balance the OH$^-$ ions content in the fuel to allow a good compromise between the positive effect on cell performance and the negative effect on the durability. Regarding the different concepts of fuel delivery and handling, ADAFCs can be categorized as passive and active. Active systems need moving parts to feed oxidant and fuel to the cell, requiring power to operate. This type of system supply has greater costs and lower system energy density than passive systems. Passive systems use natural capillary forces, diffusion, convection (air breathing) and evaporation to achieve all processes without any additional power consumption.

Alkaline direct glycerol fuel cells (ADGFCs) are of special interest, as they can be adapted to two different modes of use, by varying the type of catalyst and operational parameters, that is, either to produce only energy, as for the most part of ADAFCs, or to produce simultaneously electricity and valuable GLY oxidation products. Many works have been addressed to ADGFCs, and the number of

papers is boosted in the last years, as can be seen in the histogram in Figure 6. In the following parts of Section 3 a detailed analysis of these two ways of ADGFC utilization is reported.

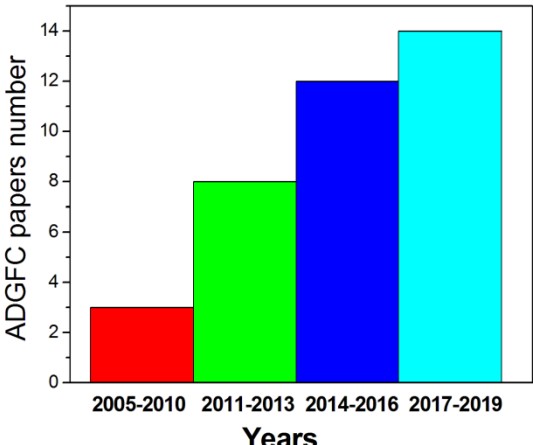

**Figure 6.** Histogram of paper numbers devoted to alkaline direct glycerol fuel cells (ADGFCs) per group of years.

### 3.1. ADGFCs for Energy Production

In the case of ADGFCs for energy production, a drawback is the low output power density, owing to the limited selectivity of the most part of the catalysts to oxidize GLY to either $CO_2$ (full oxidation) or to deeply oxidized carboxylates at the potentials of interest for fuel cell operation.

Matsuoka et al. [122] were the first to study an ADGFC, with PtRu/C and Pt/C as the anode and cathode catalyst, respectively, obtaining a maximum power density (MPD) of ca. 7 mW cm$^{-2}$ at 50 °C. Then, Bambagioni et al. [123] assembled a Pt-free ADGFC, with Pd/CNT and Fe-Co as the anode and cathode catalyst, respectively, obtaining an MPD of ca. 6 mW cm$^{-2}$ in a passive fuel cell and an MPD of ca. 80 mW cm$^{-2}$ in an active fuel cell operating at 80 °C. Later, a lot of works focused on ADGFCs for energy production [13,23,33,41,44,46,47,54–56,66,122–144]. Fuel cell performance not only depends on the catalysts and AAEM, but also on the fuel cell operating parameters, such as fuel and alkali concentration, temperature and fuel cell operation mode (active or passive). The performance, in term of maximum power density (MPD), and operating conditions of ADGFCs with different anode and cathode catalysts and AAEMs are reported in Table 4. Carbon supported Pt- and Pd-based catalysts were the most used anode materials, while Pt/C and nonprecious $N_4$-chelates of Fe and Cu/Co were the most used cathode materials. We mentioned the use of binary plasmonic Ag–Au nanoparticles modified anodes in an ADGFC: a higher power density was observed in the presence of light than that in dark condition [141]. The most used AAEMs were Tokuyama A201 and A901 membranes. Tokuyama AAEMs are hydrocarbon polymers, formed by a hydrocarbon main chain and quaternary ammonium salts, with a thickness of 28 and 10 μm for A201 and A901 membranes, respectively. The use of thin membranes is necessary to reduce ionomer resistances and to enhance water transport to the cathode catalyst sites. A drawback regarding the use of thin membranes is their high alcohol permeability, causing alcohol crossover during fuel cell operation. PTFE thin films are stable up to 260 °C, can sustain concentrated acids and bases, and permit diffusive transport through their pores while preventing mixing of fuel and oxidant streams. The performance of ADGFCs with PTFE thin film and Tokuyama A901 as the membrane was investigated for comparison [133]. Although the real performance of the ADGFC with PTFE thin film was 22.6% lower than that with A901 membrane, the performance from IR-corrected polarization curves of both ADGFCs were very close. Indeed, considering the thickness, the internal resistance of PTFE thin film (215 μm) is twice than A901 membrane (10 μm). To overcome the drawbacks (the cost and crossover issues) associated with the use of conventional ion exchange membranes, membraneless laminar flow micro fuel cells (LFμFCs) are an effective option. LFμFCs

operate by using an anode and a cathode, as it is for a conventional fuel cell; however, they dispense the use of an ionic permeable membrane. The performance of LFμFCs with glycerol as the fuel was evaluated and the results were promising [137–140].

**Table 4.** Performance, in the term of maximum power density (MPD), of ADGFCs with different catalysts and alkaline anion-exchange membranes (AAEMs). Only temperature, GLY and MOH (M = K, Na) concentration are reported in operating conditions, but it has to be remarked that MPD depends also on other parameters such as catalyst loading, gas flow and pressure. * Membraneless fuel cell; ** photochemical fuel cell.

| Fuel | Anode | Cathode | Electrolyte Membrane | Temp. °C | MPD mW cm$^{-2}$ | Refs |
|---|---|---|---|---|---|---|
| 1 M GLY/1 M KOH | PtRu/C | Pt/C | Koei Chemical Co., M.W. ca. 102,000 4-VP | 50 | 6.7 | [122] |
| 5 wt% Gly/2 M KOH | Pd/CNT | Fe-Co HypermecTM K14 | Tokuyama A-006 | 20–22, 25, 40, 60, 80 | 6 (passive ADGFC), 16, 35, 55,78 | [123] |
| 1 M GLY/2–6 M NaOH, 2 M Gly/2–6 M NaOH, 3 M Gly/2–6 M NaOH, 1 M Gly/4 M NaOH, 1 M Gly/4 M NaOH | Pt/C, Pt$_9$Bi$_1$/C, Pd/C, Pd$_9$Bi$_1$/C, Pt$_5$Pd$_5$/C | Pt/C | ADP® Solvay, Fumapem® FAA Fumatech | 25 | 8–11, 9–7, 4–0, 8, 10, 5.5, 6, 8.5 | [124] |
| 1 M Gly/2 M KOH | Pt/C | Fe-Cu-N$_4$/C HypermecTM | Tokuyama A201 | 50, 80 | 59, 125 | [125] |
| 1 M Gly/2 M KOH | Au/C | Fe-Cu-N$_4$/C HypermecTM | Tokuyama A201 | 50, 60, 70, 80 | 18, 26, 37, 58 | [126] |
| 1 M Gly/2 M KOH | Pt/C, Pd/C, Au/C | HypermecTM (Fe-Cu-N$_4$/C, Acta) | Tokuyama A201 | 50, 80, 80, 50, 80, 80, 50, 80, 80 | 59, 125, 121 (crude Gly), 38, 72, 61 (crude Gly), 18, 58, 31 (crude Gly) | [33] |
| 5 wt% Gly/2 M KOH | Pd(NiZn)/C | Fe-Co/C | Tokuyama A-201 | 25, 80 | 17 (passive ADGFC), 119 | [23] |
| 5 wt% Gly/10 wt% KOH | Pd(DBA)$_2$ | Fe-Co HypermecTM K14 | Tokuyama A-006 | 25 | 24 (passive ADGFC) | [127] |
| 3 M Gly/6 M KOH | PtCo/CNT, Pt/CNT | Fe-Cu-N$_4$/C HypermecTM | Tokuyama A901 | 80 | 285, 269 (crude Gly), 229 (crude Gly) | [54] |
| 1 M Gly/8 M KOH | Au/C | Fe-Cu-N$_4$/C HypermecTM | Tokuyama A901 | 60 | 45 | [128] |
| 1 M Gly/2 M KOH | Pt/C | Ag/C, nanocapsule, Ag/C | Tokuyama A-201 | 80 | 86, 66 (crude Gly), 45, 38 (crude Gly) | [129] |
| 2 M Gly/2 M KOH | Pd/C, Pd$_1$Au$_1$C, Pd$_1$Sn$_1$/C, Pd$_5$Au$_4$Sn$_1$/C | Pd/C | Fumasep-FAA3-PEEK | 85 (max. values) | 34, 28, 17, 51 | [44] |
| 1 M Gly/4 M KOH | Pt$_3$Sn/C | Pt/C | PBI/KOH | 30, 45, 60, 75 | 8, 17, 22, 34 | [130] |
| 1 M Gly/4 M KOH | Pt/C | Pt/C | PBI/KOH | 30, 45, 60, 75, 90 | 5, 8, 12, 18, 16 | [131] |
| 1 M Gly/2 M KOH | Pd$_{87}$Cu$_{13}$/C, Pd/C | Pt black | Tokuyama A-201 | 60 | 70, 40 | [66] |
| 3 M Gly/6 M KOH | Pd/CNT, PdAg/CNT | Fe-Cu-N$_4$/C HypermecTM | Tokuyama A901 | 80 | 180, 276 | [132] |

**Table 4.** *Cont.*

| Fuel | Anode | Cathode | Electrolyte Membrane | Temp. °C | MPD mW cm$^{-2}$ | Refs |
|------|-------|---------|----------------------|----------|--------------------|------|
| 1 M Gly/6 M KOH, 3 M Gly/6 M KOH | PdAg/CNT | Fe-Cu-N$_4$/C HypermecTM | PTFE (225 μm), PTFE (0.45 μm) | 80 | 130, 227 | [133] |
| 1 M Gly/2 M KOH | Pd/CNT, PdAg/CNT, PdAg3/CNT | Fe-Cu-N$_4$/C HypermecTM | Tokuyama A-201 | 60 | 54, 77, 70 | [13] |
| 1 M Gly/4 M KOH | Pt/C 20%, Pt/C 30%, Pt/C 40%, Pt/C 60% | Pt/C | PBI/KOH | 60 | 20, 23, 27, 36 | [134] |
| 1 M Gly/1 M KOH | Pt/C, Pt$_9$Cu$_1$/C, Pt$_7$Cu$_3$/C, Pt$_1$Cu$_1$/C | Pt/C | KOH treated Nafion | 90 | 8, 18, 16, 11 | [55] |
| 2 M Gly/5 M NaOH | PdAu/CNF, Pd black | Pt/C | Fumasep FAA-3-PK-130 | 25 | 7 (passive DGFC), 5.8 | [47] |
| 1 M Gly/4 M KOH | PtAgMnO/C, PtAg/C, Pt/C | Pt/C | PBI | 60, 90, 60, 90, 60, 90 | 46, 103, 34, 77, 23, 59 | [56] |
| 1 M Gly/21 M KOH | Au/C, Pt/C | Fe-Cu-N$_4$/C | Tokuyama A-201 | 50, 80, 50 | 18, 58, 60 | [135] |
| 0.1 M Gly/0.3 M KOH (micro fuel cell) | PtRu/C | Pt/C | Tokuyama A-201 | 25 | 1.01 (passiveADGFC), 0.94 (saponif. GLy), 0.86 (crude Gly) | [136] |
| 0.1 M Gly/0.3 M KOH | Cu@Pd/C | Pt/C | None | 25 | 20.4 | [137] * |
| 5 vol%Gly/0.3 M | Cu@Pd/C, Cu@Pt/C | Pt/C | None | 25 | 17.4, 23.2 | [138] * |
| 1.4 M Gly/8 M KOH | Au-plated Pt | Ag-plated Ni | None | 25 | 1.3 | [139] * |
| 0.05 M Gly/1 M KOH | Pt/C | Pt/C, 0.15% Fe Pt/C | None | 25 | 39, 54 | [140] * |
| 0.1 M Gly/0.1 M NaOH | AgAu and, TBAB-modified Nafion | Pt on CC | Nafion 212 | 25 | 7 × 10$^{-3}$ (dark), 15 × 10$^{-3}$ (light) | [141] ** |

As shown in the pie chart in Figure 7, at a working temperature of 80 °C, the MPD was in the 50–100 mW cm$^{-2}$ range for 43% of the data. As can be seen in Table 4, the highest MPD values were 285 and 276 mW cm$^{-2}$, obtained by ADGFCs with PtCo/CNT and PdAg/CNT, respectively, as the anode catalyst, Fe-Cu-N$_4$/C HypermecTM as the cathode catalyst and a Tokuyama A901 membrane, operating at 80 °C with 6 M OH$^-$ and 3 M GLY concentrations in fuel solution, and a ratio OH$^-$/GLY of 2:1 [13,54].

The influence of some fuel cell operating parameters such as glycerol and hydroxyl anions concentrations in fuel solution, and cell working temperature, on ADGFC performance have been evaluated. The dependence of the x–2 M OH$^-$ concentration maximum power density ratio (MPD$_{xMOH}$/MPD$_{2MOH}$) of ADGFCs on OH$^-$ concentration in fuel solutions with 1 M GLY from different datasets [13,54,124,125,128] is shown in Figure 8a. As can be seen in Figure 8a, the MPD$_{xMOH}$/MPD$_{2MOH}$ ratio increases with increasing hydroxyl anions concentration up to an OH$^-$ concentration of 8 M, where it reaches a near constant value. A higher base concentration promotes the dehydrogenation of glycerol by H$\alpha$ abstraction, leading to alkoxide formation, an active precursor to aldehyde formation.

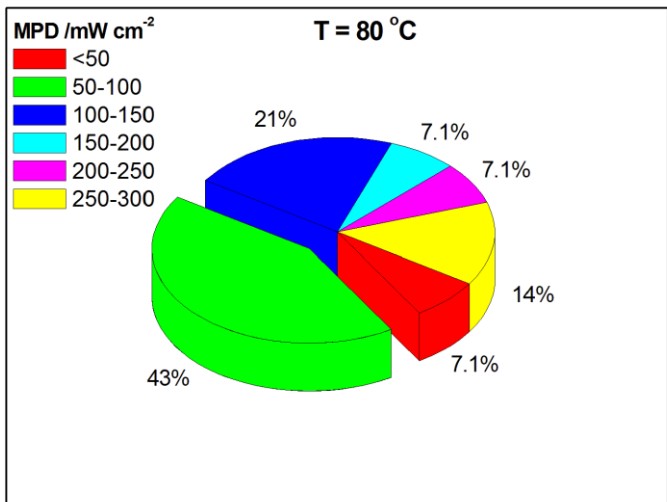

**Figure 7.** Pie chart of the MPD of ADGFCs operating at 80 °C.

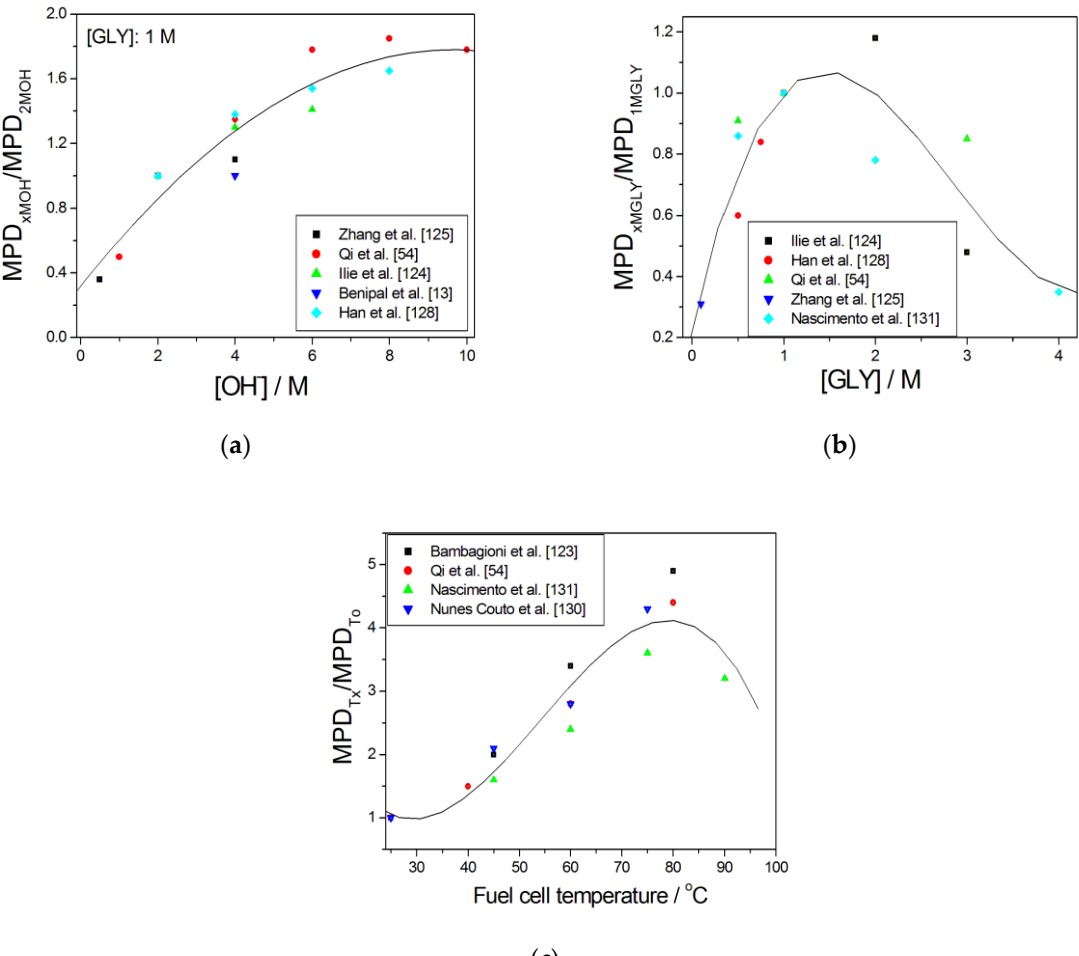

**Figure 8.** Dependence of the maximum power density of ADGFCs on some operational parameters: (**a**) [OH⁻], (**b**) [GLY] and (**c**) temperature.

Moreover, high KOH concentrations can neutralize the organic acid generated during the glycerol oxidation process, maintaining the local OH⁻ concentration to a level adequate for continuous glycerol oxidation. On the other hand, the adsorption of hydroxyl anion and glycerol is competitive. Too much

hydroxyl anion will prevent GLY adsorption on anode active sites. So, the near constant MPD values for OH$^-$ values > 8 M OH$^-$ can be explained by these simultaneous counteracting effects of OH$^-$. The dependence of the x–1 M GLY concentration maximum power density ratio (MPD$_{xMGLY}$/MPD$_{1MGLY}$) of ADGFCs on glycerol concentration in fuel solution from different datasets [54,124,125,128,131] is shown in Figure 8b. As can be seen in Figure 8b, the dependence of the MPD$_{xMGLY}$/MPD$_{1MGLY}$ ratio on GLY concentration goes through a maximum at ca. 1.5 M GLY. A similar trend was observed by Tseng and Scott for membraneless ADGFC with a maximum MPD at 1.4 M GLY [139]. It is interesting to denote that, with only one exception, all the values of the (MPD$_{xMGLY}$/MPD$_{1MGLY}$) ratio are <1. The decay of ADGFC performance for GLY concentration >1.5 M has to be mainly ascribed to the increase of the solution viscosity (pure glycerol has a high kinematic viscosity of 1490 cp at 20 °C) with increasing GLY concentration, which can limit the transport of the fuel through the diffusion layer toward the active layer [54,124]. Moreover, as previously reported for HO$^-$, the competitive adsorption between hydroxyl anion and glycerol plays an important role in determining ADFC performance. Indeed, the amount of OH$^-$ adsorbed on the active sites will decrease with increasing GLY concentration [34,54]. Thus, in addition to the OH$^-$ and GLY concentration, the [OH$^-$]/[GLY] ratio seems to be a key factor to obtain a high ADGFC performance: in light of the results obtained [13,54], an [OH$^-$]/[GLY] ratio of 2:1 seems to be the most effective. The dependence of the $x$ °C (Tx) to 25 °C (To) temperature maximum power density ratio (MPD$_{Tx}$/MPD$_{To}$) of ADGFCs on glycerol concentration in fuel solution from different datasets [54,123,130,131] is shown in Figure 8c. As can be seen in Figure 8c, the MPD$_{Tx}$/MPD$_{To}$ ratio increases with temperature up to ca. 80 °C, then a decrease of the performance was observed. A decrease of the ADGFC performance for T > 80 °C was observed also by Geraldes et al. [44]. An increase in temperature improves the electrochemical kinetics of both glycerol oxidation and oxygen reduction reactions. The conductivity of the hydroxyl ions also increases with increasing temperature, reducing the ohmic loss. Moreover, both glycerol and oxygen transport diffusivities are increased with an increase in temperature, resulting in low mass transport. However, above 80 °C, the water vapor partial pressure becomes too high, making more difficult the water balance necessary to guarantee an efficient oxygen reduction reaction. Moreover, the higher water vapor partial pressure may contribute to membrane drying, leading to a decrease in conductivity. This could explain the decrease in the cell performance at high temperature.

As previously reported, crude GLY is more than four times cheaper than refined GLY, but ADGFC tests reported in Table 4 were generally carried out using high purity glycerol as the fuel. In some cases, however, the performance of ADGFC fueled with refined GLY was compared to that ADGFC fuel with crude GLY [33,54,129,136]. Zhang et al. [33] compared the performance of ADGFCs with different anode catalysts (Pt/C, Pd/C and Au/C) fueled with pure GLY to that of the corresponding ADGFC fueled with crude GLY. The results indicated a high tolerance of the Pt/C anode against the contamination/poisoning of the impurities in crude glycerol, followed by Pd/C, while the tolerance of the impurities in crude glycerol of Au/C was considerably lower. Indeed, the performance of the cells with Pt/C, Pd/C and Au/C fueled with crude GLY was 3%, 15% and 47% lower, respectively, than that of the cells fueled with pure GLY.

The performance, in term of MPD, of ADAFCs fueled with glycerol and other alcohols were compared with each other in many works [23,66,122,123,126,132,137,142,143]. No general trend in the performance order was observed. Indeed, the order of the performance depends on various factors, intrinsic, such as the type of catalyst, and operational, such as temperature and fuel concentration. Generally, ADGFCs showed a higher performance than other types of ADAFCs at temperatures >60 °C [23,123,126,132] and low fuel concentration [143]. Regarding the effect of the type of catalyst, for example, in the same working conditions the MPD of an ADAFC with Pd-Cu/C as the anode fueled with glycerol was higher than that of the same ADAFC fueled with ethylene glycol, but the MPD of an ADAFC with Pd/C as the anode catalyst fueled with glycerol was lower than that of the same ADAFC fueled with ethylene glycol [66].

### 3.2. ADGFC for Value-Added Chemical and Energy Production

As previously reported, the partial (and selective) electrooxidation of glycerol provides a feasible way to obtain valuable chemicals of industrial interest, such as tartronate and mesoxalate. These compounds are widely used in medicine, TA (1536 USD $g^{-1}$) for the treatment of osteoporosis [145], and MOA (140 USD $g^{-1}$) for the treatment of diabetes [146] and as an inhibitor of HIV agent [147]. So, research efforts have been addressed to simultaneous generation of electricity and valuable chemicals by using an ADGFC [13,125,126,134,135,148,149]. To evaluate the product selectivity and power generation the fuel was continuously looped into the anode, at a constant cell voltage (commonly between 0.1 and 0.7 V) for a certain time [125,135,148]. Generally, the current and the power density decreases with time due to GLY concentration decrease, while the selectivity changes little during the test. [125,148]. The product selectivity of the partial glycerol electro-oxidation is significantly affected by the type of electrocatalyst, in particular, the use of gold as anode catalyst in ADGFCs for TA and MOA production and energy generation seems to be very promising [126,135,148]. Under a mild cell voltage (0.3–0.7 V), that is, in the voltage range of fuel cell operation, Au facilitated the obtaining of deeper oxidized C3 compounds, such as TA and MOA, rather than C–C bond cleavage. It has to be remarked that the degree of glycerol oxidation can be tuned by the cell potential. At fixed catalyst loading and operation conditions, the maximum amount of TA was obtained at a cell potential of 0.5 V, while the maximum amount of MOA was obtained at a cell potential of 0.3 V [135].

The product selectivity and power density of some ADGFC in different conditions is reported in Table 5. The catalyst loading also affects the TA and MOA selectivity: as can be seen in Table 5 and Figure 9, using Au/C as the anode catalyst, the highest TA selectivity was obtained for a catalyst loading of 1 $mg_{Au}$ $cm^{-2}$, while the highest MOA selectivity for a catalyst loading of 5 $mg_{Au}$ $cm^{-2}$. As can be seen in the histogram in Figure 9, at a fixed cell potential (0.3 V) the selectivity of different C3 products depends on both the anode catalyst type and the catalyst loading. At the same catalyst loading, for Pt/C the main product is GLA, but for Au/C is TA, while for the same Au/C catalyst the main product is TA for a catalyst loading of 1 $mg_{Au}$ $cm^{-2}$ and MOA for a catalyst loading of 5 $mg_{Au}$ $cm^{-2}$. An opposite trend of GLA and MOA selectivity is observed going from Pt/C to Au/C at a fixed catalyst loading (1 $mg$ $cm^{-2}$) and then increasing Au/C loading from 1 to 5 $mg$ $cm^{-2}$.

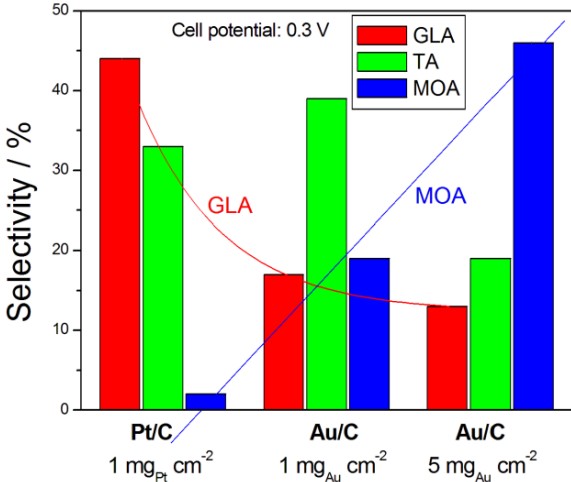

**Figure 9.** Histogram of the selectivity of ADGFCs with different anode catalysts.

**Table 5.** Selectivity, in terms of total C3 (C2) content and of the compound present in the highest concentration, and power density of some ADGFC for value-added chemical and energy production.

| Anode Catalyst | Fuel | Temp. °C | Cell Potential V | Selectivity % | Power Density mW cm$^{-2}$ | Refs |
|---|---|---|---|---|---|---|
| Au/C-NC (1 mg$_{Au}$ cm$^{-2}$) | 1 M Gly/2 M KOH | 50 | 0.5 | 75 C3, 49 TA | 2 (t = 0 h) | [126] |
| | | | 0.3 | 75 C3, 39 TA | 10 | |
| | | | 0.1 | 78 C3, 37 TA | 15 | |
| Au/C-NC (1 mg$_{Au}$ cm$^{-2}$) Au/C-AQ (1 mg$_{Au}$ cm$^{-2}$) | 1 M Gly/8 M KOH | 60 | 0.1 | 89 C3, 69 TA 85 C3, 64 TA | 15 (t = 0 h) 20 | [148] |
| Au/C-NC (5 mg$_{Au}$ cm$^{-2}$) | 1 M Gly/2 M KOH | 50 | 0.5 | 73 C3, 32 TA | 6 (t = 0 h) | [135] |
| | | | 0.3 | 78 C3, 46 MOA | 23 | |
| | | | 0.1 | 70 C3, 34 MOA | 14 | |
| Pt/C (1 mg$_{Pt}$ cm$^{-2}$) | 1 M Gly/2 M KOH | 50 | 0.7 | 84 C3, 47 GLA | 5 (averaged 2 h) | [125] |
| | | | 0.5 | 81 C3, 41 GLA | 25 | |
| | | | 0.3 | 79 C3, 44 GLA | 48 | |
| | | | 0.1 | 70 C3, 34 GLA | 33 | |
| | 1 M Gly/4 M KOH | | 0.7 | 83 C3, 46 GLA | 5 | |
| | | | 0.5 | 87 C3, 45 TA | 27 | |
| | | | 0.3 | 85 C3, 42 TA | 58 | |
| | 1 M Gly/0.5 MKOH | | 0.7 | 78 C3, 44 GLA | 1 | |
| | | | 0.5 | 71 C3, 38 GLA | 9 | |
| | | | 0.3 | 70 C3, 49 GLA | 21 | |
| | 0.1 M Gly/2 M KOH | | 0.7 | 91 C3, 50 TA | 1 | |
| | | | 0.5 | 76 C3, 40 GLA | 8 | |
| | | | 0.3 | 62 C3, 34 TA | 17 | |
| Pt/C (20 wt%; 2 mg$_{Pt}$ cm$^{-2}$) | 1 M Gly/4 M KOH | 60 | 0.4 | 80 C3, 75 TA | 20 | [134] |
| | | 90 | 0.5 | 72 C3, 67 TA | 24 | |
| | | 90 | 0.3 | 63 C3, 60 TA | 30 | |
| Pd/CNT (3 mg$_{Pt}$ cm$^{-2}$) PdAg/CNT (3 mg$_{Pt}$ cm$^{-2}$) PdAg$_3$/CNT(3 mg$_{Pt}$ cm$^{-2}$) | 1 M Gly/4 M KOH | 60 | 0.1 0.1 0.1 | 66 C3, 40 TA 57 C2, 36 OA 77 C2, 39 OA | 12 (t = 0) 12 12 | [13] |

As previously reported in Section 2, the addition of Ag to Pd facilitates the C–C bond cleavage, thus modifying the selectivity of Pd. The product selectivity of ADGFCs with Pd/C and PdAg/C as anode catalysts was investigated by Benipal et al. [13]. As can be seen in Table 5, at a cell voltage of 0.1 V, by addition of Ag to Pd, while the power density remains constant, the main product of GLY oxidation changes from TA to OA.

The OH$^-$ concentration in the fuel also affects the product selectivity of ADGFCs. As can be seen in Table 5 and Figure 10a, from data by Zhang et al. [126], on Pt/C, at fixed cell potentials (0.3 and 0.5 V) and GLY concentration, C3 selectivity increases with increasing KOH concentration. The power density also increases with KOH concentration, particularly from 0.5 to 2 M (Table 5 and Figure 10b). TA selectivity and GLY conversion also increases with increasing OH concentration: the TA/GLA ratio linearly increases with increasing KOH (Figure 10c), while GLY conversion increases with KOH concentration overall from 0.5 to 2 M, then the amount GLY conversion is almost constant (Figure 10d). The almost linear relation between power density and GLY conversion (Figure 10e) suggests that the increase of power density with increasing [OH$^-$] is essentially due to the increase of GLY conversion, rather than the amount of deeper oxidized C3 compounds. The increase of C3 selectivity, instead, is not correlated to the amount of GLY conversion, but the high enhancement of C3 selectivity going from 2 to 4 M OH$^-$, that is, for a low increase of GLY conversion, has to be ascribed to a reduced rate of C3 to C2 oxidation. As in these fuel cell working conditions [126] the rate of GLA to GA oxidation is likely higher than that of TA to GA oxidation, the enhanced rate of GLA to TA oxidation going from 2 to 4 M OH$^-$, increasing the TA/GLA ratio (Figure 10c), results in a lower amount of GA produced, and hence a higher C3 selectivity. Obviously, the GLA to TA oxidation does not change the C3 selectivity. The almost linear relation between C3 selectivity and the TA/GLA ratio (Figure 10f) confirmed this assumption.

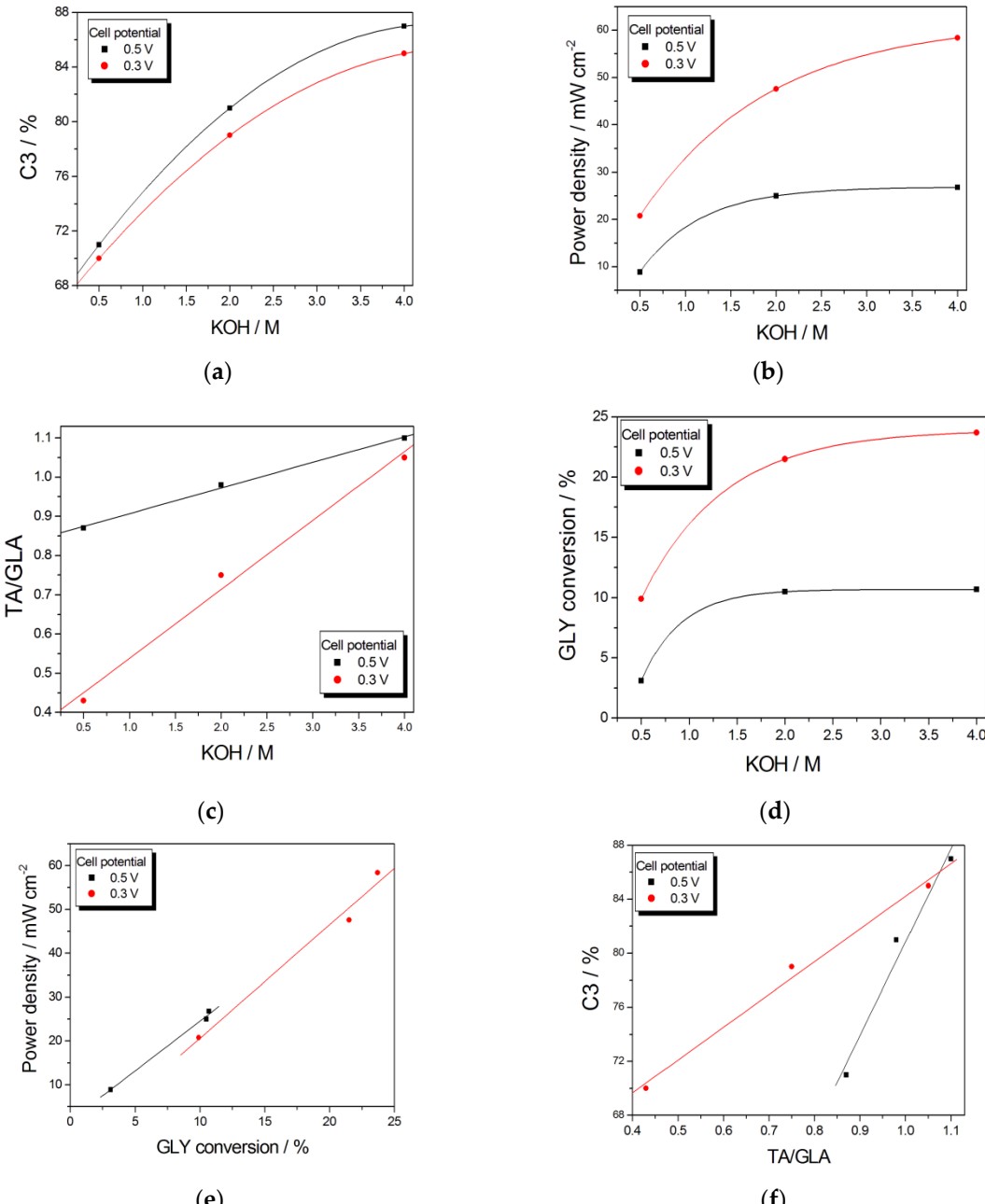

**Figure 10.** Dependence of C3 selectivity (**a**), power density (**b**), TA/GLA ratio (**c**) and GLY conversion (**d**) on [KOH] in the fuel; dependence of power density on GLY conversion (**e**), dependence of C3 selectivity on TA/GLA ratio (**f**).

Summarizing, the research on alkaline direct glycerol fuel cell overall focused on only power production. Given the great interest in obtaining valuable products, research efforts should focus on improving the performance of ADGFCs for simultaneous electricity and valuable chemical production. Moreover, so far tests on ADGFCS were almost always carried out using high purity glycerol. Considering the similar performance of ADFCs fueled with crude and high purity glycerol using Pt as the anode catalyst, due to the considerable lower cost of crude GLY, fuel cell tests should be carried out using crude GLY as the fuel.

## 4. Conclusions

This work presents an overview of the electro-oxidation of glycerol in alkaline media and its application in ADGFCs for only energy generation and for both energy and valuable chemicals production. By the comparison of the electrochemical activity and product formation of different electrocatalysts, the purpose is to provide a tool to outline new highly performing materials, leading either to a high GLY conversion and a complete glycerol oxidation to $CO_2$ or to direct the selectivity to obtaining desired compounds. Precious metals, such as Pt, Pd and Au, are the most active catalysts for GLY electro-oxidation, but on one hand the high cost of Pt and Pd and their not fully satisfying electrochemical stability, and on the other hand the high overpotentials of Au are drawbacks for their use in ADGFCs. Bifunctional catalytic materials play an important role in glycerol electro-oxidation, affecting reaction activity and product formation pathways. The addition of a second/third metal to precious catalysts improves both the activity and the selectivity, either to C3 or to C2 compounds, by hindering or promoting C–C bond cleavage, respectively. Due to their low cost, the use of non-noble metal catalysts, such as Ni- and Cu-base materials, has been investigated for the GOR. A high selectivity to C2 and C1, in particular to glycolic acid and formic acid, are frequently observed for glycerol oxidation under basic conditions with non-noble metal catalyst systems. Their GOR activity, however, is not satisfactory.

For ADGFC for energy production, to maximize GLY oxidation, catalysts able to break the C–C bond, such as PtRh and PtAg, should be used, while for ADGFC for energy and valuable C3 products, catalysts, which suppress C–C bond cleavage, such as PtBi and PtSn, possibly without reducing electricity production, that is, by increasing GLY conversion, should be employed. In addition to the search for suitable catalysts, an evaluation of the effect of operation conditions, such as $OH^-$ and GLY concentration and cell temperature, on power generation has been made. Optimal values of $OH^-$ and GLY concentration and cell temperature have been reported. While a lot of works were addressed to ADGFCs for only energy generation, few papers focused on ADGFC for both energy and valuable compounds production. Based on the results of half-cell measurements, more tests of ADGFC for simultaneous electricity and valuable chemical production should be carried out.

The performance of ADGFCs fueled with crude and high purity glycerol showed only a little difference: considering that the average price of crude glycerol is ca. 80% lower than that of high purity glycerol, but the performance of direct crude glycerol fuel cells with Pt-based anodes is only slightly lower (ca. 5%) than that of direct high purity glycerol fuel cells, the use of crude GLY was strongly suggested.

Summarizing, there is still much room to improve ADGFC performance by developing novel low cost materials: as the actual GOR activity of non-precious materials is far from that of noble metals, future work should be addressed to the development of non-precious catalysts with novel structures, such as 1D and core-shell materials, having a catalytic activity comparable to that of precious metals. In addition, as actually ADGFC tests are carried out in different operational condition, while taking into account the type and loading of the catalyst, efforts should be focused to the optimization and standardization of the operational fuel cell conditions, by achieving suitable chemical ($OH^-$ and GLY concentration and their ratio in the fuel stream) and physical (temperature, pressure, flow rate) parameters.

**Funding:** This research received no external funding.

**Conflicts of Interest:** The author declares no conflict of interest.

## Abbreviations

| | |
|---|---|
| AAEM | alkaline anion-exchange membrane |
| ADAFC | alkaline direct alcohol fuel cell |

ADGFC      alkaline direct glycerol fuel cell
C3,C2,C1   glycerol oxidation products with 3, 2 and 1 carbon atoms, respectively
DFT        density functional theory
DHA        dihydroxyacetone
FA         formic acid
GA         glycolic acid
GALD       glyceraldehyde
GLA        glyceric acid
GLY        glycerol
GOA        glyoxylic acid
GOR        glycerol oxidation reaction
HPA        hydroxypyruvic acid
LA         lactic acid
MOA        mesoxalic acid
MPD        maximum power density
OA         oxalic acid
TA         tartronic acid

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
