# Peer review of "Glycerol Electro-Oxidation in Alkaline Media and Alkaline Direct Glycerol Fuel Cells"

_catalysts, doi:10.3390/catal9120980_

Round 1
Reviewer 1 Report
The current review article is on electro-oxidation of glycerol in alkaline media and its relevance for alkaline direct glycerol fuel cells. The author addresses many aspects highlighting the importance of research on glycerol:
green and renewable carbon source as building blocks in chemicals importance of electro-catalysts to address the activity and selectivity, and application of the oxidation products for fuel cells.The manuscript is well written and structured. I suggest to accept this manuscript as it will attract a broad community in research in industry. However, I believe few cosmetic works need to be undertaken to make this article more reader-friendly and attractive:
1.) Fig. 1 appears too messy for the readers due to inter-crossing arrows. Some re-organisation of the chemical structure is needed (incl. colours?). Use whole space for Fig. 1. Furthermore, I suggest to add number of electrons involved for each oxidation product on top of the arrows. Also the typical potential range for the accordant product should be under each arrow, if these potential ranges are available.
2.) The authors focuses on anodic oxidation as summarised in equation (1), which is the main focus of the review. I would also add the reaction product(s) with total number electrons that happens at the cathode in equation a second (2), to brief the reader about the total reaction. What product is formed at the cathode? Molecular Hydrogen? If it is hydrogen (H2), is it been collected as alternative energy carrier or exposed to the air? Please address in few sentences.
3.) Table 2: Please add in a separate row the potential ranges that appear for the selectivity (for each entry).
4.) Table 3: Also here, the reported potential range is required for each selectively generated oxidation product (for each entry).
5.) Other than that, just few spacing errors, typos and text formating need attention.
Regards
Author Response
Reviewer 1
1)The number of electrons involved for each oxidation product has been added on top of the arrows of Figs. 1 and 2. There is little space in the figure to make other additions.
2)The reaction products with total number electrons that occurs at the cathode was added as the equation (9), and the overall reaction has been added as the equation (10). At the cathode OH- anions are formed.
3) and 4) The potential ranges for the selectivity have been added in Table,1 Table 2 and Table 3.
Reviewer 2 Report
In this paper the author reviews studies of oxidation of glycerol and of its use as a fuel in glycerol fuel cells. In the introduction interest in this subject is synthesized and justified, 13 citations being added to facilitate the understanding of the potential readers. In section 2. studies of oxidation of glycerol in alkaline media are reported. In the subsection 2.1 single Pt, Pd and Au metals, binary PtAu/PdAu catalysts, and Pt-,Pd-, Au-based binary/ternary catalysts are considered.This part includes 2 tables, 5 figures and 78 citations; of these only 37 have been published in the period 2016-2019, and many are older than 20 years.In the subsection 2.2 non-precious metal catalysts are analysed. This time 1 table is included and 27 references but only 13 are from 2016-2019. Again the review does not reflect an accurate, up-to-date depiction of the state of the art in the field,particularly for non-metal electrocatalysts where advances have been made recently. Moreover, the provided discussion does not give at all a rational picture of the research carried out so far. Section 3. deals with the glycerol fuel cells; this time there are 30 citations but only 11 references are for the period 2016-2019. Two tables and 5 figures affect positively the quality of the paper, but I regret feeling unable to consider the paper as an acceptable one.
Author Response
Reviewer 2
Comment: “Moreover, the provided discussion does not give at all a rational picture of the research carried out so far.”
Answer: The research carried out so far was mainly aimed to the improvement of the catalytic activity and selectivity of noble metals by the addition of a second/third metal. Few works, instead, was addressed to the improvement of the activity for glycerol oxidation of less expensive non-precious metals, whose selectivity for C2 and C1 products is high. Therefore, much space remains for research aimed at increasing the catalytic activity of non-precious catalysts, maintaining their high selectivity.
The research on alkaline direct glycerol fuel cell overall focused on only power production. Given the great interest in obtaining valuable products, research efforts should focus on improving the performance of ADGFCs for for simultaneous electricity and valuable chemical production. Moreover, so far tests on ADGFCS were almost always carried out using high purity glycerol. Considering the similar performance of ADFCs fueled with crude and high purity glycerol using Pt as the anode catalyst, due to the considerable lower cost of crude GLY, fuel cell tests should be carried out using crude GLY as the fuel.
Reviewer 3 Report
In general the paper is well methodological written and relevant references to the previous works in the field are well documented. In my opinion, the article may be accept for publication in the present form.
Author Response
I thank the reviewer for the positive comment.